# Chemotherapy and Radiotherapy Long-Term Adverse Effects on Oral Health of Childhood Cancer Survivors: A Systematic Review and Meta-Analysis

**DOI:** 10.3390/cancers16010110

**Published:** 2023-12-25

**Authors:** Joana Pombo Lopes, Inês Rodrigues, Vanessa Machado, João Botelho, Luísa Bandeira Lopes

**Affiliations:** 1Clinical Research Unit, Egas Moniz Center for Interdisciplinary Research (CiiEM), Egas Moniz School of Health & Science, 2829-511 Almada, Portugal; joanampmlopes@gmail.com (J.P.L.); ines.rodrigues1042000@gmail.com (I.R.); vmachado@egasmoniz.edu.pt (V.M.); jbotelho@egasmoniz.edu.pt (J.B.); 2Evidence-Based Hub, Egas Moniz Center for Interdisciplinary Research (CiiEM), Egas Moniz School of Health & Science, 2829-511 Almada, Portugal

**Keywords:** chemotherapy, radiotherapy, children, oral health, late side effects

## Abstract

**Simple Summary:**

The survival rate for pediatric cancer has increased over the past few decades, short- and long-term complications have been detected and studied, and oral complications have emerged as an important topic of research. Here, we aimed to highlight the importance of oral manifestations that may only become apparent years or even decades after cancer treatment. Childhood cancer survivors presented a higher risk of having dental alterations than control counterparts. Additional analyses reveal possible sex-based differences that should be explored in future studies. These results collectively highlight the importance of oral healthcare and the prevention of disease in childhood cancer survivors.

**Abstract:**

The survival rate for pediatric cancer has increased over the past few decades, short- and long-term complications have been detected and studied, and oral complications have emerged as an important topic of research. Here, we aimed to highlight the importance of oral manifestations that may only become apparent years or even decades after cancer treatment. This systematic review was conducted according to the Preferred Reporting Items for Systematic Reviews and Meta-Analysis. We searched articles using PubMed via the MEDLINE, Web of Science, and LILACS databases until October 2023. Overall, 35 observational studies were included, and the results estimated a pooled prevalence of the following dental anomalies: discoloration, 53%; crown-root malformations and agenesis, 36%; enamel hypoplasia, 32%; root development alterations, 29%; unerupted teeth, 24%; microdontia, 16%; hypodontia, 13%; and macrodontia, 7%. Most childhood cancer survivors have at least one dental sequela. Childhood cancer survivors presented a higher risk of having dental alterations than control counterparts. Additional analyses reveal possible sex-based differences that should be explored in future studies. These results collectively highlight the importance of oral healthcare and the prevention of disease in childhood cancer survivors.

## 1. Introduction

Childhood cancer is a leading cause of death, with an estimated 400,000 children and adolescents between the ages of 0 and 19 years diagnosed with cancer [1,2]. Because they are generally not prevented or detected by screening, accurate and timely diagnosis is essential to promote clinical success and high survival rates [2]. The treatment options for pediatric malignancies include chemotherapy, radiation therapy, surgery, and multimodal approaches [1,2].

The survival rates for children with cancer have increased; nevertheless, up to 40% of children present complications later due to cancer treatment [3]. Short- and long-term complications have been identified, and oral complications are an important research topic [3]. In addition, children are three-times more likely than adults to experience developmental complications, exacerbating the impact of searching for this topic [3]. Some oral manifestations may occur early during treatment or years or decades after cancer treatment. Short-term adverse effects may include dental caries, mucositis, bleeding, taste alterations, secondary infections, periodontal disease, trismus and osteoradionecrosis [3,4]. Long-term complications were not described until the 1970s because the post-treatment observation period was still short [5,6]. More recently, combined anticancer treatments have been identified as being responsible for late oral effects, including craniofacial and dental developmental defects and salivary gland dysfunction, especially when performed at a young age [3,7,8,9].

This systematic review aimed to summarize the findings of estimating the prevalence of oral short- and long-term adverse effects in pediatric cancer survivors during and after oncologic treatment. We aimed to provide information that will allow for the reinforcement of the role of pediatric oncologists for possible dental abnormalities that have a negative impact on the quality of life of both patients and families. 

## 2. Materials and Methods

### 2.1. Protocol and Registration

All authors established the protocol, registered it at the National Institute for Health Research PROSPERO platform (ID Number: CRD42022336369), and reported it according to the Preferred Reporting Items for Systematic Reviews and Meta-Analysis (PRISMA) checklist [10] (Appendix A). 

### 2.2. Focused Questions and Eligibility Criteria

We developed a protocol to answer two PICO questions: “What is the prevalence of late oral health adverse effects in childhood cancer survivors with a history of chemotherapy and radiotherapy”?“Are children who undergo cancer therapy more likely to have late oral health adverse effects when compared with healthy controls counterparts”?

Late oral health adverse effects were defined as late sequelae of oncological treatment-related toxicities to dentofacial structures.

The respective statements were as follows: pediatric patients with malignant cancer diagnosed between the ages of 3 and 18 years (P, Participants); patients who had undergone a therapeutic combination of radiotherapy/chemotherapy or not by the age of 18 years and were in the primary/mixed/permanent dentition were included (I, Intervention); the presence or absence of a control group was not a limitation (C, Control); estimated prevalence of the late effects of the oral complications (mucositis, candidiasis, ulcers) and dental structures (microdontia, hypodontia, hypoplasia, malformed teeth, impaired root growth, interrupted root growth, V-shaped roots, taurodontism, premature apical closure, and tooth agenesis) (O, Outcome).

Randomized clinical trials, controlled clinical trials, cohort studies (prospective or retrospective design), and cross-sectional studies were eligible for inclusion. The exclusion criteria were as follows: (1) duplicate studies; (2) abstracts, commentaries, reviews, letters to the editor, consensus, opinions, case studies, and case series; (3) unpublished information; (4) lack of appropriate clinical measures; (5) secondary analysis of data sourced from a previous study; and (6) inclusion of animal studies. There were no restrictions on the year or language of publication.

### 2.3. Data Search Strategy and Study Selection

We searched PubMed through MEDLINE, Web of Science, and LILACS for all relevant articles published until October 2023. Grey literature was also searched for using OpenGrey (http://www.opengrey.eu/, accessed on 20 November 2023). The following search terms were used: (1) (chemotherap* OR radiotherap* OR cancer); (2) (child* OR adolescent* OR pediat* OR paediat*); (3) (caries OR decay OR xerostomia OR root stunting OR periodont* OR gum OR gingiv*) NOT adult*. Two independent reviewers (J.P.L. and L.B.L.) performed the search and included studies.

Two independent examiners performed, in duplicate, the assessment of titles and/or abstracts of retrieved studies independently (J.P.L. and L.B.L.). For measurement reproducibility, inter-examiner reliability following full-text assessment was calculated using kappa statistics. Any disagreements were resolved by discussion with a third author (M. M.).

### 2.4. Data Extraction Process and Data Items

Data extraction was performed by two reviewers independently and in duplicate (J.P.L. and L.B.L.). Any paper deemed potentially eligible by one of the reviewers was independently reviewed. All disagreements were resolved by discussion with a third reviewer (VM). The following information was collected: general description, research characteristics, methodology, and outcome measures. The following standard information was extracted from each eligible study: the first author’s name, year of publication, country and place of sampling, study period, sample size (male/female), case definition setting, observation setting, sampling strategy, cancer type, treatment (chemotherapy and/or radiotherapy), adverse oral health effects, study funding, and risk of bias.

### 2.5. Risk of Bias (RoB) Assessment

The methodological quality of the eligible studies was assessed using the Joanna Briggs Institute (JBI) Critical Appraisal Checklist [11]. This tool allowed for analysis in eight domains, presented in the form of questions as follows: (1) Were the criteria for inclusion in the sample clearly defined? (2) Were the study participants and settings described in detail? (3) Was exposure measured in a valid and reliable way? (4) Were objective and standard criteria used for the measurement of the condition? (5) Were confounding factors identified? (6) Were the strategies to deal with the confounding factors stated? (7) Were the outcomes measured in a valid and reliable manner? (8) Was appropriate statistical analysis used? Each item was scored as Y (i.e., yes)—reported and adequate, N (i.e., no)—not reported, and U (i.e., unclear)—reported inadequately. Any disagreements between examiners were resolved through discussion with a third author. Only studies with all items scored with “Y” were considered to be of high quality, studies with at least one item “N” were of low quality, and, finally, for those which presented at least one “U” item and all the others “Y” were of unclear quality. The Risk-Of-Bias VISualization (ROBVIS) tool was used to analyze the risk of bias [12].

### 2.6. Summary Measures and Synthesis of Results

Standard spreadsheet software (Microsoft Excel for Mac, version 16.50. Microsoft, Redmond, WA, USA) was used for data extraction. Frequencies and percentages were used to describe categorical variables, whereas continuous variables were reported as mean ± standard deviation (SD) and range. Random-effects meta-analysis and forest plots of prevalence were calculated in R version 3.4.1 (R Studio Team 2018) using the ‘meta’ package [13], through the DerSimonian–Laird random-effects meta-analysis. A meta-analysis was performed to calculate dental anomalies in pediatric cancer survivors. A risk ratio (RR) with a 95% confidence interval (CI) was used to describe the dental disharmonies of cancer survivors compared to healthy children. The RR was pooled using a random-effects model in R version 3.4.1 (R Studio Team 2018), using the ‘readxl’ package and pairwise random-effects meta-analysis, and *p*-values less than 0.05 were considered statistically significant. The chi-square (χ^2^) test was used to calculate overall homogeneity, and substantial heterogeneity was considered when I^2^ statistics exceeded 50% [14]. To explore potential sources of heterogeneity, we performed a subgroup analysis according to the methodological quality of the included studies and the female/male ratio. Publication bias was considered when the meta-analysis included at least 10 studies [14].

## 3. Results

### 3.1. Study Selection

The online search strategy identified 3601 potentially relevant publications. After removing duplicates, 3029 articles were assessed against the eligibility criteria, and 2950 were excluded after title and/or abstract review. Of the 79 articles assessed for eligibility for full-paper review, 44 were excluded, with the respective reasons for exclusion detailed in Appendix A. As a result, a final number of 35 observational studies were included for qualitative synthesis; a PRISMA diagram is shown in Figure 1. The inter-examiner reliability of the full-text screening was considered very high (kappa score = 0.915, 95% CI: 0.895–0.925).

### 3.2. Studies’ Characteristics

Overall, a total of 3761 participants from all 35 included studies were included in this systematic review, 2625 childhood cancer survivors (889 females and 1122 males, 10 did not report sex) and 1136 healthy children (209 females and 243 males, 11 did not report sex) (Table 1). All studies addressed long-term adverse oral health effects in childhood cancer survivors, although 14 studies did not present a control group [4,15,16,17,18,19,20,21,22,23,24,25].

Regarding the type of study, 15 were cohort studies [17,18,19,20,21,22,23,24,26,27,28,29,30,31,32], 13 case–control studies [33,34,35,36,37,38,39,40,41,42,43,44,45], and 7 cross-sectional studies [3,4,15,16,25,46,47].

Several points were considered in the case definition setting. Some studies addressed multiple topics: 24 assessed caries incidence [4,15,17,22,24,26,27,28,29,30,31,33,34,36,37,39,40,41,43,44,45,46,47,48], 18 assessed dental abnormalities such as root stunting and microdontia [4,16,18,20,21,23,26,27,29,39,40,41,43,44,46,47,48], and 15 stressed the developmental defects of enamel [16,17,21,23,26,27,29,30,36,40,41,44,46,47]. The other 13 studies mentioned plaque index and/or gingival index [3,15,17,24,26,27,28,30,31,34,37,39,44], and the other 9 considered oral hygiene [15,24,30,37,38]. Saliva assessment was addressed by seven studies [21,25,27,38,39,41,42], and two investigated craniofacial development [26,28].

Some research highlights themes in a unique way, such as the regularity of dental attendance and type of dentist visited [37], number of erupted teeth relative to age [30], oral mucositis and ulceration, candidiasis, herpes and herpetic gingivo-stomatitis, oral petechiae, facial pain [15], already [21] addressed facial asymmetry and jaw hypoplasia, as well as trismus. Hutton 2010 mentioned traumatized teeth and [35] calculated the root surface areas of mandibular teeth.

Furthermore, studies have been conducted in 16 countries worldwide. Notably, no studies have been performed in Oceania or Africa.

The most prevalent types of cancer studied were acute lymphoblastic leukemia, Hodgkin’s lymphoma, non-Hodgkin’s lymphoma, Rhabdomyosarcoma, Wilms tumor, and neuroblastoma. Other malignant conditions included Retinoblastoma, Fibroma, Medulloblastoma, Nasopharyngeal carcinoma, Langerhans cell histiocytoma, malignant teratoma, optical glioma, germinoma, leiomyosarcoma, and hepatoblastoma. Treatment modalities included chemotherapy and radiotherapy, with or without bone marrow transplantation.

**Table 1 cancers-16-00110-t001:** Characteristics of the included studies.

Study	Design	Country	Sample	Oral Health Conditions Case Definition Setting	Cancer Type	Treatment Modality	Study Funding
Halperson et al. (2022) [4]	Cross-sectional	Israel	121	Dental caries; Dental developmental anomalies (DDA—includes five major groups: no disturbance identified, hypomineralization or hypoplasia, microdontia, root changes, and an absent tooth bud categorized as hypodontia); DMFT index	leukemia\lymphoma in 53 (45%) patients, solid tumors in 35 (29%) and other hematological conditions leading to BMT in 31 (26%)	Most patients (83, 69%) had received ChemoT without radiotherapy. Thirty-eight (31%) had received radiation therapy only or in combination with ChemoT. Fourteen (12%) of the cohort had received total body irradiation (TBI) 12 Gray and 15 (13%) radiation to the head and/or neck area (range of 27–70 Gray). The remaining nine patients had received radiotherapy to other areas (range of 30–70 Gray). Thirty percent of the cohort had undergone BMT	NR
Shayani et al. (2022) [31]	Retrospective	Spain	109	DDE, microdontia, taurodontism, agenesis, root shortening (RS)	leukemias and lymphomas (41.3%) followed by solid non-CNS tumors (38.5%) and, finally, solid CNS tumors (20.2%)	ChemoT (CT); CT combined with radiotherapy (CT + RT); and CT + RT combined with hematopoietic stem cell transplantation (HSCT)	NI
Rabassa-Blanco et al. (2022) [23]	Retrospective	Chile	23	missing or filled teeth index and the presence of gingivitis	ALL	ChemoT	NI
Stolze et al. (2022) [25]	Cross-sectional	The Netherlands	291	unstimulated (UWS) and stimulated whole salivary flow rates (SWS) were measured according to internationally standardized procedures—categorized into ‘hyposalivation’ (<0.2 mL/min and <0.7 mL/min, respectively) and ‘severe hyposalivation’ (<0.1 mL/min and <0.5 mL/min, respectively); partic- ipants were asked to fill out the Dutch translation of the Xerostomia Inventory (XI)	Hematological malignancy (*n* = 216); brain tumor (*n* = 19); solid tumor (*n* = 57)	head and neck radiotherapy (H&N RT) or total body irradiation (TBI) without chronic graft versus host disease (cGVHD), a group of CCS with (a history of) cGVHD after HSCT, and a group of CCS treated with ChemoT and no H&N RT or TBI	NI
Tanem et al. (2022) [47]	Cross-sectional	Norway	46	decayed-missing-filled index (DMFT), oral dryness, maximum mouth opening (MMO), fungal infection, and registration of dental developmental disturbances (DDD) in the form of hypodontia, microdontia, and enamel hypoplasia	brain tumors medulloblastoma (MB) and central nervous system supratentorial primitive neuroectodermal tumor (CNS-PNET).	ChemoT + craniospinal irradiation	Research Grant
Guagnano et al. (2022) [46]	Cross-sectional	Italy	52	Decayed-missing-filled teeth (dmft/DMFT) index; Disturbances of enamel mineralisation using Aine rating scale; dental age estimation using panoramic radiographs; dental abnormalities using the Höltta Defect Index on panoramic radiographs—Valores médios para cada sexo, tipo de terapêutica e idade no diagnóstico (<5 anos ou ≥5 anos), os valores apresentados à frente são média da populção toda	ALL Acute Myeloblastic Leukemia Medulloblastoma Familiar Hemophagocitic Lymphohistiocitosis Lymphoma Juvenile Myelomonocytic Leukemia Wilms tumour Epatoblastoma Rhabdomyosarcoma Ewing-PNET Sarcoma Severe Aplastic Anaemia Xantoastocitoma Wide Cells Anaplastic Lymphoma Histiocytosis	CT and/or RT, Hematopoietic Stem Cell Transplantation (HSCT) or Bone Marrow Transplantation (BMT)	NI
Seremidi et al. (2021) [32]	Retrospective	Greece	70	Microdontia, Malformed teeth, Oligodontia, Hypodontia, Enamel defects, Dental caries	central nervous system tumor, Solid Tumors and Lymphomas	ChemoT, or hemopoietic stem cell transplantatio	None
Proc et al. (2019) [3]	Cross-sectional	Poland	75	dmft; DMFT; plaque index by silness and loe	ALL; Wilms tumor; Neuroblastoma; Rhabdosarcoma (RMS); Brain tumor; Hepatoblastoma; Acute non-lymphoblastic leukemia (ANLL) Non-Hodgkin’s lymphoma (B-NHL) Hodgkin’s lymphoma (HL); Primitive neuroectodermal tumor (PNET) Germinal tumour; Tumor ovari	RadioT & ChemoT	NI
Alnuaimi et al. (2018) [15]	Retrospective	United Arab Emirates	120	Oral health problem: oral mucositis & ulceration, candidiasis, herpes and herpetic gingivo-stomatitis, gigival bleeding, gigivites, oral petechiae, dental caries, poor oral hygiene, facial pain/palsy, other	Leukaemic	ChemoT	NI
Çetiner et al. (2019) [28]	Retrospective	Turkey	53	Gingival Index, Plaque Index, dmft/DMFT, dmfs/DMFS, craniofacial development	Hodgkin lymphoma, Non-Hodgkin lymphoma, Neuroblastoma, Wilms tumor, Retinoblastoma, Rhabdomyosarcoma, Nasopharynx carcinoma	ChemoT	NI
Olczak-Kowalczyk et al. (2018) [45]	Case–control	Poland	60	DMFT; dmft; DMFS; dmfs: teeth/surfaces with white spot lesions–WSL (D1 + 2/d1 + 2), following the ICDAS-II criteria	neoplasm; medulloblastoma (12.5%), nephroblastoma (Wilms’tumour,10.8%), Burkitt’s lymphoma (10.8%), neuroblastoma (8.3%), rhab- domyosarcoma (RMS, 6.6%), Ewing’s sarcoma (5.8%), and less frequently: chondrosarcoma, hepatoblastoma, glioblas- toma, ependimoma, and osteosarcoma.	Multidrug therapy, adapted to each neoplasm type and including vincristine, cyclophosphamide, adriamycin, etopo- side, cisplatin, ifosfamide, actomycin, and methotrexate; ChemoT for the others	NI
Bica et al. (2017) [16]	Retrospective	Romania	36	hypoplasia (hypomineralisation) of the enamel, microdontia and atypical eruption.	limphoblastic leukemia	ChemoT	NI
Krasuska-Sławińska et al. (2016) [38]	Case–control	Poland	60	oral hygiene, gingiva (PI), dentition, and potential visible decrease in salivary secretion.	Different neoplasms	PCH—60 patients after at least 1 year ChemoT CG—60 generally healthy patients.	NR
Owosho et al. (2016) [21]	Retrospective	United States of America	13	Facial asymmetry and jaw hypoplasia; Effects on the dental tissue causing tooth agenesis/hypodontia, root agenesis/stunting/malformation, and/or enamel hypoplasia; trismus, hyposalivation/xerostomia.	head and neck rhabdomyosarcoma (HNRMS)—Tumor sites were orbit in 1 patient and parameningeal in 12 (infratemporal fossa in 5, nasopharynx in 5, parapharyngeal in 1, and middle ear in 1)	multiagent ChemoT and IMRT—median radiation dose to the primary tumor was 50.4 Gy (range: 45–50.4 Gy), and the ChemoT agents were vincristine, doxorubicin, cyclophosphamide, ifosfamide, and etoposide	NI
Nemeth et al. (2014) [43]	Case–control	Hungary	38	DMFT; unstimulated saliva flow rate—spitting method (USF); stimulated saliva flow rate—spitting method (SSF); palatal saliva flow rate using a Periotron meter (Oraflow Inc., Plainview, NY, USA) (PS); salivary buffer capacity using CRT buffer (Ivoclar Vivadent AG, Schaan, Lichtenstein)	NI	18 patients BFM-95 = protocol for acute lymphoblastic lymphoma, Berlin-Frankfurt-Munster; 5 patients NBL-2 = protocol for neuroblastoma; 4 patients CWS 96 = protocol of Cooperative Soft Tissue Sarcoma Study Group; 4 patients SIOP 93 = international protocol of the Interna- tional Society of Paediatric Oncology; 3 patients BFM-98 = protocol for acute lymphoblastic lymphoma, Berlin-Frankfurt-Munster; 2 patients COSS-96 = protocol of Cooperative Os- teosarcoma Study Group; 2 patients DAL-HD 90 = protocol for Hodgkins disease, No patients had radiotherapy treatment, nor bone marrow transplantation, nor stem cell transplantation	NI
Nemeth et al. (2013) [42]	Case–control	Hungary	38	DMFT; CPI; radiographic dental examination was used to analyze dental malforma- tions: agenesis, without third molars, microdontia, macrodontia, unerrupted teeth; root malformation	NI	18 patients BFM-95 = protocol for acute lymphoblastic lymphoma, Berlin-Frankfurt-Munster; 5 patients NBL-2 = protocol for neuroblastoma; 4 patients CWS 96 = protocol of Cooperative Soft Tissue Sarcoma Study Group; 4 patients SIOP 93 = international protocol of the Interna- tional Society of Paediatric Oncology; 3 patients BFM-98 = protocol for acute lymphoblastic lymphoma, Berlin-Frankfurt-Munster; 2 patients COSS-96 = protocol of Cooperative Os- teosarcoma Study Group; 2 patients DAL-HD 90 = protocol for Hodgkins disease, No patients had radiotherapy treatment, nor bone marrow transplantation, nor stem cell transplantation	NI
Lauritano et al. (2012) [29]	Prospective	Italy	52	DMFT, microdontia, enamel hypoplasia, dental agenesis, v-shaped roots	Thirty- nine patients were affected by lymphoblastic leukaemia (ALL), the remaining ones were affected by acute myeloblastic leukaemia (AML)	Patients were treated according to Italian Association of Paediatric Hematoncology (AIEOP)—Methotrexate + Vincristine + Daunoblastine + Prednisone + Desamethasone. Seven patients with ALL received cranial irradiation (18 Gy) in addition to ChemoT and cytotoxic treatment	NR
Hutton et al. (2010) [17]	Retrospective	United Kingdom	120	DMFT index; dmft index; enamel opacities, fissure sealed, microdont; traumatized teeth; basic periodontal examination and gingival bleeding score in patients with fully erupted permanent incisors and first molars	Wilm’s tumour—29 patients (24.2%), rhabdomyosarcoma—10 patients (8.3%), Hodgkin’s lymphoma—14 patients (11.7%), non-Hodgkin’s lymphoma—10 patients (8.3%), neuroblastoma—21 patients (17.5%), and other solid tumour types—36 patients (30.0%)	ChemoT—four principal groups of chemo- therapeutic agent used: high-dose chemo- therapy with stem-cell rescue (HDCSCR); anthracycline drugs; alkylating agents; platinum drugs; and overlapping regimes	NR
Maciel et. al. (2009) [39]	Case–control	Brazil	56	agenesis, microdontia, macrodontia, short roots, tapering roots, enlarged pulp chambers, supernumerary teeth, taurodontism, DMFT score, visible plaque index (VPI), gingival bleeding index (GBI), saliva flow	ALL	ChemoT, Chemo/radiotherapy, Chemo/radio/BMT	Research Grant
Çubukçu et al. (2008) [33]	Case–control	Turkey	62	DMF/T, dmf/t	Non-Hodgkin lymphoma, Retinablastoma, Hodgkin lymphoma, Fibroma, Medulloblastoma, Wilms tumor, Nasopharyngeal carcinoma, Langerhans cell histiocytoma, Neuroblastoma, Malignant teratoma, Optical glioma, Rhabdomyosarcoma, Disgerminoma, Leiomyosarcoma, Hepatoblastoma	ChemoT	NI
Avşar et al. (2007) [27]	Retrospective	Turkey	96	DMFT, The Silness-Loe Plaque Index (PI) and Gingival Index (GI), Saliva assessment included salivary flow rate, salivary buffer capacity, mutans streptococci, and lactobacilli counting, disturbances of enamel mineralization, disturbances in dental development	Hodgkin’s or non-Hodgkin’s lymphoma	ChemoT	NI
Marec-Berard et al. (2005) [40]	Case–control	France	27	microdontia, excessive caries, root stunting, hypodontia, and enamel hypoplasia	nephroblastoma	Institutional protocol (SIOP 93 protocol) consisting of poly ChemoT with vincristine, actinomycin ± doxorubicin without any head and/or neck ir- radiation or high-dose ChemoT	NR
Oguz et al. (2004) [44]	Case–control	Turkey	36	DMFT; DMFS; Loe–Silness GI; Sillnes–Loe PI; enamel defects and discolorations; root malformations; eruption status; agenesis; premature apexifications and microdontia	non- Hodgkin’s lymphomas (NHL)	Twenty-seven patients were treated according to BFM-90 B cell protocol; while the LSA2 L2 protocol was used in 4 patients, and the LMT-89 protocol was administered in five patients	NI
Duggal et al. (2003) [35]	Case–control	United Kingdom	69	Calculation of root surface areas of mandibular teeth	Acute lymphoblastic leukaemia (43.3%); Wilms tumor (14.4%), Hodgkin’s disease (9.3%); CNS tumors (8.2%) Non Hodgkins lymphoma, acute myeloid leukaemia and other diagnoses	ChemoT, radiotherapy, and both chemo-and prophylactic cranial irradiation of between 16 and 22GY, or had received fractionated total body irradiation and a bone marrow transplant	NR
Pajari et al. (2001) [22]	Retrospective	Finland	36	DMFT	18 suffering from leukemia and 18 from solid tumors	combination ChemoT and 4 patients also received cranial irradiation	NI
Alpaslan et al. (1999) [26]	Retrospective	NI	32	discoloration, enamel hypoplasia, crown/root malformation, unerupted teeth, premature apexification, microdontia, agenesis, gingival and plaque indexes, denatal caries, craniofacial growth	Hodgkin’s or non-Hodgkin’s lymphoma	ChemoT	NI
Kaste et al. (1998) [20]	Retrospective	United States of America	52	dental abnormalities	Neuroblastoma	8 received head and/or neck irradiation, either as part of a preparative regimen for bone marrow transplantation (n= 2) or as local therapy of a metastasis (n = 6)	NR
Duggal et al. (1997) [36]	Case–control	United Kingdom	46	Enamel defects—modified developmental defects of enamel index (DDE index); DMFTS index; avaliação gengival	22 acute lymphoblastic leukaemia; 6 Hodgkins disease; 4 Non- Hodgkins lymphoma; 6 brain tumours, 4 Wilm’s tumour; 4 other childhood malignancies.	Multi-drug ChemoT with or without cranial irradiation	NR
Kaste et al. (1997) [18]	Retrospective	United States of America	423	Dental abnormalities: root stunting (abnormally shortened roots), microdontia (abnormallly small teeth), or hypodontia (absent teeth)	ALL	Multiagent ChemoT; In addition, cranial irradiation (1800 or 2400 cGy) was given to 243 of the 423 children (55.6%).	NR
Kaste et al. (1995) [19]	Retrospective	United States of America	22	Dental abnormalities: root stunting, microdontia and hypodontia; multiple abnormalities.	Head and neck rhabdomyosarcoma	Multiagent ChemoT (including cyclophosphamide, Adriamy- cin, vincristine, and dactinomycin) and radiotherapy on four successive treatment regimens	NR
Sonis et al. (1995) [24]	Case–control	Belgium	52	DMFT; dmft: Gengival index; Plaque index	27 acute lymphoblastic leukaemia; 7 non-Hodgkin’s lymphoma; 7 Wilms’ tumour; 5 rhabdomyosarcoma; 6 different childhood cancers	ChemoT. Patients had not received any radiotherapy to the oral or the salivary gland region	NI
Dens et al. (1995) [34]	Retrospective	NI	64	dmft; DMFT; OHI-S; modified loe and silness gingival index score	ALL	Varied combinations of chemotherapeutic agents: ChemoT alone (group 1); 1800 cGy (group 2); 2400 cGy (group 3)	NI
Näsman et al. (1994) [41]	Case–control	Sweden	76	Dental caries, salivary flow, salivary microbial counts, enamel disturbances, and disturbances in dental development	BMT group: 15 children were treated for acute leukemia, 1 for a B-cell lymphoma,3 for Gaucher’s disease, 1 for a severe combined immunodeficiency. ChT group: 21 were treated for acute leukemia, 9 for lymphoma,6 for Wilm’s tumor, 6 for rhabdomyosarcoma,3 for histiocytosis-X, 3 for neuroblastoma, 3 for optic glioma, 3 for other CNS-tumors, and 3 for other tumors	Bone marrow transplantation (BMT group); ChemoT	NR
Fleming et al. (1993) [37]	Case–control	Northern Ireland	54	Regularity of dental attendance; type of dentist visited; toothbrushing frequency; plaque presence on buccal and lingual surfaces; gengivitis (através do sangramento gengival ao passar com a sonda); DMFT index; dmft index	ALL	ChemoT	NR
Purdell-Lewis et al. (1988) [30]	Cohort	United Kingdom	45	oral hygiene index; papilllary bleeding index; number of erupted teeth relative to age; number of carious or filled primary and permanent teeth; percentage of primary teeth with initial lesions; percentage of erupted incisors, canines or permanent first molars with opacities (1), rough surfaces (2), vertical grooves (3), hypoplastic horizontal grooves and pits scored using DDE-index	acute lymphatic leukaemia, neuroblastoma, wilm’s tumor, rhabdomyosarcoma, Histiocytosis X, acute non-lymphatic leukemia	poly ChemoT	NI

NI—No information; NR—Not reported; RadioT—Radiotherapy; ChemoT—Chemotherapy; ALL—Acute limphoblastic leukaemia; DDE—Developmental defects of enamel (DDE).

### 3.3. Methodological Quality of the Included Studies

Most studies were categorized with high methodological quality (*n* = 21, 60%), while six had unclear methodological quality, and eight were of low methodological quality (Table 2).

Studies mostly failed on stating strategies to deal with confounding factors (60.0%, *n* = 21) (item 6), clearly defining criteria for sample inclusion (54.3%, *n* = 19) (item 1), identifying confounding factors (31.4%, *n* = 11) (item 5), and describing, in detail, study subjects and the setting (22.9%, *n* = 8) (item 2). The remaining items had a performance of over 95%.

### 3.4. Data Synthesis

#### 3.4.1. Dental Anomalies Prevalence

We were able to estimate specific prevalence rates for eight dental anomalies (Table 3) (forest plots are available in Appendix A). Discoloration was the most prevalent dental anomaly, with a mean prevalence of 53% (0.53, 95% CI: 0.42; 0.65), but was less described (only in four studies) [17,26,28,44]. The second dental anomalies most prevalent, with 36% prevalence, were crown-root malformations (0.36, 95% CI: 0.28; 0.44) and agenesis (0.36, 95% CI: 0.27; 0.45), both described in 10 studies [16,21,23,26,27,28,29,39,42,44], followed by 32% prevalence of enamel hypoplasia (0.32, 95% CI: 0.21; 0.45), described in 13 studies [4,16,20,21,26,27,28,29,30,39,40,41,44]. With 29% (0.29, 95% CI: 0.16; 0.43) prevalence, root development alterations were described in 10 studies [4,21,23,26,27,28,29,39,40,44]. Unerupted teeth had a mean prevalence of 24% (0.24, 95% CI: 0.15; 0.34), and this condition was described in four studies [26,28,42,44], with microdontia at 16% (0.16, 95% CI: 0.09; 0.24), but this was the most commonly described dental anomaly in 14 studies [4,17,18,19,20,23,26,27,28,29,39,40,42,44]. Lastly, hypodontia, reported in six studies [4,15,18,19,20], had a prevalence of 13% (0.13, 95% CI: 0.05; 0.23), and macrodontia was the least prevalent dental anomaly, with 7% (0.07, 95% CI: 0.04; 0.12), being described in only 5 five studies [18,27,39].

Furthermore, studies have been conducted in 16 countries worldwide. Notably, no studies have been performed in Oceania or Africa. 

#### 3.4.2. Dental Anomalies Risk in Pediatric Cancer Patient Survivors Compared to Controls

When comparing the prevalence of dental anomalies between cancer survivors and controls, five out of eight were significantly more prevalent among survivors (Table 4) (forest plots are available in Appendix A). Root development alterations were 591% (OR = 6.91, 95% CI: 3.89; 12.29) more commonly found in survivors than in controls; microdontia was 518% (OR = 6.18, 95% CI: 2.45; 15.56), discoloration was 468% (OR = 5.68, 95% CI: 3.02; 10.7), agenesis was 350% (OR = 3.50, 95% CI: 1.98; 6.16), and enamel hypoplasia was 95% (OR = 1.95, 95% CI: 1.32, 2.88). The prevalence of crown-root malformation, unerupted teeth, and macrodontia was not significantly different between cancer survivors and controls.

Other oral manifestations, extending from dental development anomalies to soft tissue and saliva alterations, were reported in the studies but were not included in the meta-analysis due to a lack of data amenable for it; nevertheless, alternative synthesis methods were used.

No statistically significant differences were found on craniofacial growth among the controls and cancer survivors [26,28].

Hyposalivation in childhood cancer survivors is relatively high [25,41,43], with more significant alterations found in stimulated salivary flow [27,43]. Ref. [39] was the only study reporting no alterations in saliva flow rates. Studies did not find alterations in salivary buffer capacity [41,43], but a salivary microbial flora shift in patients who received radiation therapy was found, with an increased number of mutans streptococci and lactobacilli in saliva [27,41].

Caries experience was assessed through the calculation of decayed–missing–filled teeth and surfaces for primary (dmft, dmfs) and permanent (DMFT, DMFS) dentition. Permanent dentition scores (DMFT and DMFS) were, for the majority of the studies, higher in cancer survivors when compared with controls [3,27,31,37,38,42,45]). Only one study reported a higher caries level in primary dentition [17], and the others reported no differences between groups [24,36,39].

### 3.5. Additional Analysis

We further assessed, through sensitivity analyses, whether the risk of bias (Table 5) and the female–male ratio (Table 6) could interfere with the estimates. Risk of bias only proved to be significant in the root development alteration (*p* < 0.0001). 

Female–male ratio showed a significant effect in the estimates concerning root development alteration (*p* < 0.0001), enamel hypoplasia (*p* = 0.0001), discoloration (*p* = 0.047), and microdontia (*p* = 0.0204), unveiling a possible sex-based difference.

## 4. Discussion

### 4.1. Summary of Main Findings

The results of the present systematic review estimated the pooled prevalence of the following dental anomalies as long-term dental sequelae in patients who had undergone cancer therapy during early childhood: discoloration, 53%; crown-root malformations and agenesis, 36%; enamel hypoplasia, 32%; root development alterations, 29%; unerupted teeth, 24%; microdontia, 16%; hypodontia, 13%; and macrodontia, 7%. Compared with controls, these dental anomalies were significantly more prevalent in cancer survivors and pediatric patients. Root development alterations were 591%, microdontia was 518%, discoloration was 468%, agenesis was 349%, and enamel hypoplasia was 95% more likely to be found in cancer survivors than in controls.

### 4.2. Implications for Practice and Research

As previously mentioned, the late side effects of chemotherapy and radiotherapy on the stomatognathic system in pediatric cancer survivors are numerous, which challenges clinical care and management in the dental setting. Regarding cancer types, it is perceived that the most prevalent cancers in children worldwide are leukemias, with the highest rate, followed by tumors of the central nervous system, then lymphomas, and others. [4,16,23]. Thus, most patients receive chemotherapy without radiotherapy, but they may receive radiotherapy alone or in combination with chemotherapy. Radiation therapy to the head and/or neck area can range from 27 to 70 Gray [4]. And we also know that dental development or odontogenesis is a complex process that occurs over a long period of time, starting in intrauterine life and ending at 14–15 years of age [4,16]. Thus, each tooth goes through different stages of development, which when subjected to extrinsic or intrinsic factors, can result in the appearance of dental development defects. Depending on the stage of odontogenesis that is affected, different changes may occur; that is, if any changes occur during histodifferentiation, the structure of enamel and dentin may be altered. In turn, if they occur during morphodifferentiation, they may cause shape and size abnormalities of the teeth, and if the disturbances persist, they can damage root formation, resulting in a shortened or tapered root, which, in turn, can impair tooth eruption and occlusion. The first signs are expected after one to two years of anticancer treatment [4]. Some antineoplastics inhibit odontogenesis and eruption and can induce qualitative and quantitative changes in dental tissues. Regarding radiotherapy treatments, exposure to radiotherapy doses greater than 20 Gy has been shown to contribute to a greater risk of developing dental anomalies [28].

Therefore, alterations in root development, microdontia, discoloration, agenesis, and enamel hypoplasia, which were the most common alterations recorded, had an impact on the quality of life.

With all this in mind, it is our understanding that, given the possibility of the presence of dental abnormality and increased caries risk as a consequence of cancer treatment, the most acceptable course of action should be to assume that the quality of life and oral implications are real, so the normal dental therapy scenario may increase the level of clinical priority for preventive screening and early screening.

Despite all the included oral manifestations, crown malformation, prevalence of unerupted teeth, and macrodontia, craniofacial growth was not statistically significant between controls and cancer survivors. However, a higher level of caries in the primary dentition has only been reported once [17], as well as alterations in saliva flow rates [39].

### 4.3. Strengths and Limitations

This study was conducted following PRISMA, a strict and widely advised guideline that has increased robustness and decreased reporting errors. Furthermore, a comprehensive literature search was conducted using a meticulous predefined protocol. Nevertheless, there are some limitations that need to be discussed. It is possible to see that there are several studies that address late health effects; however, in a non-systematic way and on multiple distinct points, this leads to a low sample size of children with cancer, where it is essential to obtain consistent results.

Dental abnormalities have been addressed, but studies on saliva alterations are scarce and have different objectives, making them inconclusive, [25,27,39,41,43], as well as and despite reports of a higher prevalence of caries in the permanent dentition [3,27,31,37,38,42,45]. In the deciduous dentition, the results showed that there were no differences between the groups [24,36,39], with only one reporting the opposite [17]. One point that was not mentioned was malocclusion and occlusal disharmonies, which would be interesting given the high prevalence of changes in number and tooth development.

Thus, future studies should focus on data representativeness and method standardization to ensure more homogeneous evidence-based results. This information is extremely relevant to pediatric oncologists and to raise awareness among oral health professionals regarding the possible and predictable problems they are facing.

## 5. Conclusions

Childhood cancer survivors presented a higher risk of having dental alterations than control counterparts. Also, this group of people also presents considerable prevalence of such conditions. Additional analyses reveal possible sex-based differences that should be explored in future studies, as well as more longitudinal studies, as this is the only way to assess and understand the oral consequences of antineoplastic agents. These results collectively highlight the importance of oral healthcare and the prevention of disease in childhood cancer survivors. 

## Figures and Tables

**Figure 1 cancers-16-00110-f001:**
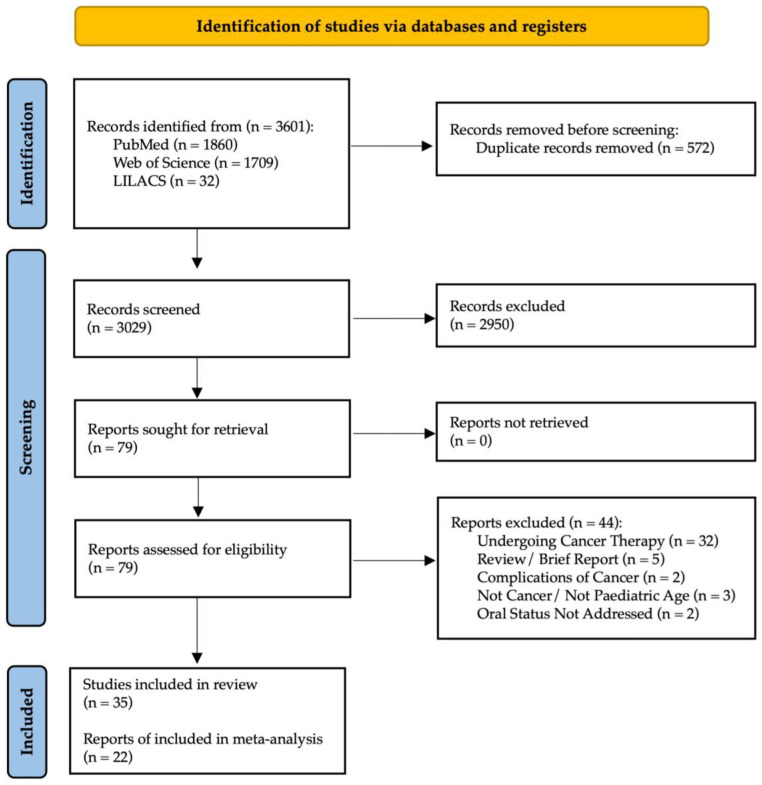
PRISMA flowchart of studies.

**Table 2 cancers-16-00110-t002:** Results from the methodological appraisal using JBI Critical Appraisal Checklist.

Study	1	2	3	4	5	6	7	8	Overall
Halperson et al. (2022) [4]	U	Y	Y	Y	Y	N	Y	Y	High
Shayani et al. (2022) [31]	Y	N	Y	Y	Y	N	Y	Y	High
Rabassa-Blanco et al. (2022) [23]	Y	N	Y	Y	N	N	Y	Y	High
Stolze et al. (2022) [25]	Y	Y	Y	Y	N	N	Y	Y	High
Tanem et al. (2022) [47]	Y	Y	Y	Y	Y	N	Y	Y	High
Guagnano et al. (2022) [46]	N	N	Y	Y	N	N	Y	Y	High
Seremidi et al. (2021) [32]	Y	Y	Y	Y	Y	Y	Y	Y	Low
Proc et al. (2019) [3]	U	Y	Y	Y	Y	Y	Y	Y	Unclear
Alnuaimi et al. (2018) [15]	U	N	U	Y	Y	N	Y	Y	High
Çetiner et al. (2019) [28]	U	Y	Y	Y	Y	Y	Y	Y	Unclear
Olczak-Kowalczyk et al. (2018) [45]	Y	Y	Y	Y	Y	N	Y	Y	High
Bica et al. (2017) [16]	Y	U	Y	Y	N	N	Y	Y	High
Krasuska-Sławińska et al. (2016) [38]	U	Y	Y	Y	Y	N	Y	Y	High
Owosho et al. (2016) [21]	Y	Y	Y	Y	Y	Y	Y	Y	Low
Nemeth et al. (2014) [43]	Y	Y	Y	Y	Y	N	N	Y	High
Nemeth et al. (2013) [42]	Y	Y	Y	Y	Y	Y	Y	Y	Low
Lauritano et al. (2012) [29]	U	Y	Y	Y	Y	Y	Y	Y	Unclear
Hutton et al. (2010) [17]	U	Y	Y	Y	N	N	Y	Y	High
Maciel et. al. (2009) [39]	U	Y	Y	Y	N	N	Y	Y	High
Çubukçu et al. (2008) [33]	U	Y	Y	Y	N	N	Y	Y	High
Avşar et al. (2007) [27]	Y	Y	Y	Y	Y	Y	Y	Y	Low
Marec-Berard et al. (2005) [40]	U	Y	Y	Y	N	N	Y	Y	High
Oguz et al. (2004) [44]	U	Y	Y	Y	Y	Y	Y	Y	Unclear
Duggal et al. (2003) [35]	Y	Y	Y	Y	Y	Y	Y	Y	Low
Pajari et al. (2001) [22]	U	Y	N	N	Y	N	Y	Y	High
Alpaslan et al. (1999) [26]	U	Y	Y	Y	Y	N	Y	Y	High
Kaste et al. (1998) [20]	U	N	Y	Y	Y	N	Y	Y	High
Duggal et al. (1997) [36]	Y	Y	Y	Y	N	N	Y	Y	High
Kaste et al. (1997) [18]	U	N	Y	Y	N	N	Y	Y	High
Kaste et al. (1995) [19]	U	N	Y	Y	N	N	Y	Y	High
Sonis et al. (1995) [24]	U	Y	Y	Y	Y	Y	Y	Y	Unclear
Dens et al. (1995) [34]	U	Y	Y	Y	Y	Y	Y	Y	Unclear
Näsman et al. (1994) [41]	Y	Y	Y	Y	Y	Y	Y	Y	Low
Fleming et al. (1993) [37]	Y	Y	Y	Y	Y	Y	Y	Y	Low
Purdell-Lewis et al. (1988) [30]	Y	Y	Y	Y	Y	Y	Y	Y	Low

Y—Yes; U—Unclear; N—No. Items description: 1—Were the criteria for inclusion in the sample clearly defined?; 2—Were the study subjects and the setting described in detail?; 3—Was the exposure measured in a valid and reliable way?; 4—Were objective, standard criteria used for measurement of the condition?; 5—Were confounding factors identified?; 6—Were strategies to deal with confounding factors stated?; 7—Were the outcomes measured in a valid and reliable way?; 8—Was appropriate statistical analysis used?

**Table 3 cancers-16-00110-t003:** Prevalence data on dental anomalies in pediatric cancer patient survivors.

Clinical Alteration	Studies (*n*)	Cases (*n*)	Effect	I^2^	*p*-Value	Egger Test
Root development alteration	10	595	0.29 (0.16; 0.43)	92	<0.0001	0.6154
Crown-root malformation	11	1052	0.31 (0.20; 0.44)	92	<0.0001	0.4814
Unerupted teeth	4	159	0.24 (0.15; 0.34)	49	0.1176	-
Enamel hypoplasia	13	695	0.32 (0.21; 0.45)	91	<0.0001	0.1060
Hypodontia	6	765	0.13(0.05; 0.23)	89	<0.0001	-
Discoloration	4	241	0.53 (0.42; 0.65)	64	0.0397	-
Agenesis	10	521	0.36 (0.27; 0.45)	77	<0.0001	0.0677
Microdontia	14	1237	0.16 (0.09; 0.24)	91	<0.0001	0.6624
Macrodontia	5	722	0.07 (0.04; 0.12)	71	0.0077	-

**Table 4 cancers-16-00110-t004:** Risk ratio on dental anomalies in cancer survivor pediatric patients.

Clinical Alteration	Studies (n)	Cases/Controls (n/n)	Effect	I^2^	*p*-Value	Egger Test
Root development alteration	5	272/260	6.91 (3.89; 12.29)	0	0.4406	6.91 (3.89; 12.29)
Crown-root malformation	5	269/244	1.60 (0.32; 7.98)	95	<0.0001	1.61 (0.24; 10.61)
Unerupted teeth	3	121/96	1.50 (0.62; 3.60)	40	0.1877	1.50 (0.62; 3.60)
Enamel hypoplasia	7	401/310	1.95 (1.32; 2.88)	0	0.6990	1.95 (1.32; 2.88)
Discoloration	3	121/96	5.68 (3.02; 10.7)	0	0.6825	5.68 (3.02; 10.7)
Agenesis	8	415/392	3.50 (1.98; 6.16)	52	0.0333	3.50 (1.98; 6.19)
Microdontia	7	362/352	9.49 (3.13; 28.70)	22	0.2983	9.13 (3.17; 26.30)
Macrodontia	3	190/192	1.90 (0.60; 5.99)	0	0.5527	1.90 (0.60; 5.99)

**Table 5 cancers-16-00110-t005:** Sensitivity analysis of risk of bias on prevalence using meta-regressions.

Sensitivity Analysis	Studies (*n*)	Cases (*n*)	Effect	I^2^ (%)	*p*-Value
Root development alteration					
Low ROB	2	109	0.59 (0.49; 0.68)	0	<0.0001
High or Unclear ROB	8	486	0.22 (0.11; 0.35)	90	
Crown-root malformation					
Low ROB	2	134	0.40 (0.18; 0.63)	84	0.7064
High or Unclear ROB	9	848	0.35 (0.26; 0.44)	82	
Unerupted teeth					
Low ROB	1	38	0.15 (0.06; 0.29)	-	0.1848
High or Unclear ROB	3	121	0.27 (0.17; 0.39)	48	
Enamel hypoplasia					
Low ROB	4	230	0.27 (0.04; 0.60)	96	0.6724
High or Unclear ROB	9	465	0.35 (0.23; 0.47)	87	
Low ROB	3	147	0.38 (0.16; 0.62)	85	0.8598
High or Unclear ROB	7	374	0.35 (0.26; 0.46)	76	
Microdontia					
Low ROB	2	134	0.16 (0.00; 0.48)	93	0.9748
High or Unclear ROB	12	1103	0.16 (0.09; 0.25)	91	
Macrodontia					
Low ROB	2	134	0.04 (0.01; 0.09)	0	0.2247
High or Unclear ROB	3	722	0.10 (0.03; 0.18)	84	

ROB—Risk of bias.

**Table 6 cancers-16-00110-t006:** Sensitivity analysis of female and male ratio on prevalence data on dental anomalies in cancer survivor pediatric patients.

Clinical Alteration	Estimate	95% CI	*p*-Value
Root development alteration	−0.16	−0.25; −0.07	0.0004
Crown-root malformation	0.01	−0.08; 0.09	0.8895
Unerupted teeth	0.04	−0.09; 0.16	0.5626
Enamel hypoplasia	0.13	0.07; 0.20	0.0001
Hypodontia	−0.15	−0.50; 0.20	0.3901
Discoloration	0.08	0.02; 0.13	0.0047
Agenesis	0.04	−0.05; 0.13	0.3825
Microdontia	−0.09	−0.16; −0.01	0.0204
Macrodontia	−0.01	−0.41; 0.39	0.9632

## Data Availability

Data were extracted from the original studies included in this systematic review. All data used for statistical analyses are presented in the manuscript and its Appendix A.

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
