# Peer review of "Chemotherapy and Radiotherapy Long-Term Adverse Effects on Oral Health of Childhood Cancer Survivors: A Systematic Review and Meta-Analysis"

_cancers, 2023, doi:10.3390/cancers16010110_

Round 1

Reviewer 1 Report

Comments and Suggestions for Authors

This article studied the effects of chemotherapy and radiotherapy on oral tissue in childhood cancer. It addresses a fundamental issue, as the side effects of childhood cancer are becoming more and more significant as treatment outcomes improve.

Introduction: I don't think there are any particular problems.

Materials and Methods: I don't think there are any particular problems.

Results: I don't think there are any particular problems.

Discussion: Many of the problems that can occur are tooth-related. The impact of chemotherapy and radiotherapy during tooth development. It would be better to describe what mechanisms can affect the development of the teeth. I would also like you to describe at what age treatment is most common or affects the teeth.

Author Response

We are pleased with the opportunity to revise and submit our manuscript entitled “Chemotherapy and radiotherapy long-term adverse effects on oral health 
of childhood cancer survivors: a systematic review and meta-analysis” (Manuscript ID - cancers-2776218).

Considering the editor and reviewers’ comments, all have been considered very important and were taken into profound consideration.

Manuscript changes are highlighted in the revised manuscript. Our point-by-point responses to all comments are outlined and detailed below. We hope that you find our responses satisfying.

We hope the revised manuscript will enable its further consideration. We are happy to consider further revisions and we thank you for your continued interest in our research.

 Dear Reviewer

Thank you for all your valid comments, I hope I have clarified it in the best possible way, and I am available for any other questions that arise.

Discussion: Many of the problems that can occur are tooth-related. The impact of chemotherapy and radiotherapy during tooth development. It would be better to describe what mechanisms can affect the development of the teeth. I would also like you to describe at what age treatment is most common or affects the teeth.

Our Answer: Indeed, our results support your claim, that most problems found are tooth-related. Aiming to better describe which mechanisms can affect odontogenesis, we have improved the Discussion section, which we believe now addresses the impact of chemotherapy and radiotherapy on such development. The following sentence was added and reads as follows: “And we also know that dental development or odontogenesis is a complex process that occurs over a long period of time, starting in intrauterine life and ending at 14-15 years of age (Halperson 2022, Bica 2017). Thus, each tooth goes through different stages of development, which when subjected to extrinsic or intrinsic factors, can result in the appearance of dental development defects. Depending on the stage of odontogenesis that is affected, different changes may occur; that is, if any changes occur during histodifferentiation, the structure of enamel and dentin may be altered, and, if they occur during morphodifferentiation they may cause shape and size abnormalities of the teeth. If the disturbances persist, root formation can damage, resulting in a shortened or tapered root, which in turn can impair tooth eruption and occlusion. The initial signs are expected after one-two years of anticancer treatment (Halperson 2022). Some antineoplastics agents inhibit odontogenesis and eruption, and can induce qualitative and quantitative changes in dental tissues. Exposure to radiotherapy doses greater than 20 Gy has been shown to contribute to a greater risk of developing dental anomalies (Cetiner 2018)”.

Reviewer 2 Report

Comments and Suggestions for Authors

This systematic review aimed to summarize the findings of estimating the prevalence of oral short- and long-term adverse effects in pediatric cancer survivors during and after oncologic treatment

Strengths:

Comprehensive Review: The article provides an extensive overview of long-term oral health effects in childhood cancer survivors, covering a wide range of dental anomalies.

Importance of Oral Health: Highlights the critical need for specialized dental care in this patient group, emphasizing an often-overlooked aspect of survivorship.

Methodological Rigor: The systematic approach and meta-analysis strengthen the validity of the findings.

Weaknesses:

Limited Scope on Cancer Types and Treatments: The study could benefit from a broader examination of various cancer types and treatment modalities.

Need for Longitudinal Studies: The article could be strengthened by including or recommending more longitudinal studies to understand the long-term trajectory of oral health issues.

Lack of Practical Guidelines: While the article stresses the need for specialized care, it falls short in providing practical guidelines or strategies for dental practitioners.

 the meta-analysis graphics in the article, it would be advisable to:

 ·       Use Clear, Comprehensive Visuals: Graphs should be easy to interpret and visually clear, illustrating the statistical outcomes of the meta-analysis effectively.

·       Incorporate Forest Plots: These are essential for meta-analyses, showing the individual study results, the weight of each study, and the overall effect size.

·       Include Heterogeneity Measures: Graphs representing the variability among study results, such as I² statistics, would be beneficial.

·       Present Subgroup Analyses: If applicable, graphs that detail the effects within different subgroups (e.g., types of cancer or age groups) would provide deeper insights.

·       Ensure Accessibility: Graphs should be accessible, with proper labeling and legends, to cater to a broad range of readers, including those not specialized in the field.

 Author Response

We are pleased with the opportunity to revise and submit our manuscript entitled “Chemotherapy and radiotherapy long-term adverse effects on oral health 
of childhood cancer survivors: a systematic review and meta-analysis” (Manuscript ID - cancers-2776218).

Considering the editor and reviewers’ comments, all have been considered very important and were taken into profound consideration.

Manuscript changes are highlighted in the revised manuscript. Our point-by-point responses to all comments are outlined and detailed below. We hope that you find our responses satisfying.

We hope the revised manuscript will enable its further consideration. We are happy to consider further revisions and we thank you for your continued interest in our research.

Dear Reviewer

Thank you for all your valid comments, I hope I have clarified it in the best possible way, and I am available for any other questions that arise.

Weaknesses:

  1. Limited Scope on Cancer Types and Treatments: The study could benefit from a broader examination of various cancer types and treatment modalities.

Our answer: We followed your recommendation accordingly. We added: Regarding, cancers types, it’s perceived that the most prevalent cancers in children worldwide are leukemias with the highest rate, followed by tumors of the central nervous system, then lymphomas, and others. (Rabassa blanco 2022, Bica 2107, Halperson 2022). Thus, most patients receive chemotherapy without radiotherapy, but they may receive radiotherapy alone or in combination with chemotherapy. Radiation therapy to the head and/or neck area can range from 27 to 70 Gray. (Halperson 2022). 

2.     Need for Longitudinal Studies: The article could be strengthened by including or recommending more longitudinal studies to understand the long-term trajectory of oral health issues.

Our answer: We thank this commentary and we fully agree, We added: “, as it is the only way to assess and understand the oral consequences of antineoplastic agents”

  1. Lack of Practical Guidelines: While the article stresses the need for specialized care, it falls short in providing practical guidelines or strategies for dental practitioners.

Our answer:  Although we understand this suggestion, the goal of this systematic review was not to develop  practical guidelines or strategies for dental practitioners. To meet such standard we would have to employ others strategies and guidelines such as the AGREE guideline.

  1. In the meta-analysis graphics in the article, it would be advisable to:

Our answer: We appreciate this suggestion. We have made them available as supplementary files. Due to an informatic lapse, they were not in the platform.

Reviewer 3 Report

Comments and Suggestions for Authors

Dear authors,

Thank you for the opportunity to review your manuscript “Chemotherapy and radiotherapy long-term adverse effects on oral health of childhood cancer survivors: a systematic review and meta-analysis”. This review evaluates an important topic regarding the oral health complications that arise following cancer therapy in childhood survivors. I was pleased to read your thorough synthesis and meta-analysis of the existing evidence. However, there are a few issues in the paper that should be addressed before it can be considered for publication in Cancers.

Main issues:

  1. Introduction
  • The introduction provides helpful context and outlines the research questions well. Consider summarizing or omitting some background details to allow more space for detailing the rationale and importance of the specific research questions.
  1. Methods
  • The methods generally follow PRISMA guidelines but the search strategy could be described in more detail. Consider reporting the full electronic search strategy for one database as an example in the supplementary material.
  • Please clarify the inclusion/exclusion timeframe as currently there are no restrictions stated regarding publication year.
  1. Results
  • The results section clearly lays out the main findings but is lacking some key information. Please add a PRISMA flow diagram detailing the study screening and selection process.
  • Please include a Table summarizing the key characteristics of all included studies.
  • There are some discrepancies between the number of studies mentioned in the text (n=35) vs the number actually referenced. Please reconcile.
  1. Discussion
  • The discussion interprets the main results well. However, the implications for practice could be expanded and the limitations require more detail. Specifically, evaluate the impact of potential sources of bias, how generalizable the results are, how the evidence might translate clinically, and what key next research steps are to advance this field of research further.
  1. Writing quality
  • There are frequent grammatical errors throughout that must be addressed with careful editing by a native English speaker. This will improve the clarity and readability greatly.

I appreciate the attention to an important research area regarding the long-term adverse effects of childhood cancer treatment. With major revisions addressing the comments outlined, I believe this systematic review and meta-analysis may provide a helpful synthesis of the current understanding that could inform clinical practice and future research directions.

Best regards,

Comments on the Quality of English Language

Moderate editing of English language required

Author Response

We are pleased with the opportunity to revise and submit our manuscript entitled “Chemotherapy and radiotherapy long-term adverse effects on oral health 
of childhood cancer survivors: a systematic review and meta-analysis” (Manuscript ID - cancers-2776218).

Considering the editor and reviewers’ comments, all have been considered very important and were taken into profound consideration.

Manuscript changes are highlighted in the revised manuscript. Our point-by-point responses to all comments are outlined and detailed below. We hope that you find our responses satisfying.

We hope the revised manuscript will enable its further consideration. We are happy to consider further revisions and we thank you for your continued interest in our research.

Dear Reviewer

Thank you for all your valid comments, I hope I have clarified it in the best possible way, and I am available for any other questions that arise.

  1. The introduction provides helpful context and outlines the research questions well. Consider summarizing or omitting some background details to allow more space for detailing the rationale and importance of the specific research questions.

Our answer: We have not understood this suggestion. We ask a further clarification on this point to better attend your expectations.

  1. Methods
  • The methods generally follow PRISMA guidelines but the search strategy could be described in more detail. Consider reporting the full electronic search strategy for one database as an example in the supplementary material.

Our answer: We have made available the detailed syntax used for this study.

  • Please clarify the inclusion/exclusion timeframe as currently there are no restrictions stated regarding publication year.

Our answer: there is no restriction on the publication year.

  1. Results
  • The results section clearly lays out the main findings but is lacking some key information. Please add a PRISMA flow diagram detailing the study screening and selection process.

Our answer: we appreciate your comment, but the PRISMA flow diagram was submitted initially.

  • Please include a Table summarizing the key characteristics of all included studies

Our answer: thank you, and added

  • There are some discrepancies between the number of studies mentioned in the text (n=35) vs the number actually referenced. Please reconcile.

Our answer: Thank you for pointing this out. Checked

  1. Discussion
  • The discussion interprets the main results well. However, the implications for practice could be expanded and the limitations require more detail. Specifically, evaluate the impact of potential sources of bias, how generalizable the results are, how the evidence might translate clinically, and what key next research steps are to advance this field of research further.

Our answer: We appreciate your comment. We followed your recommendations of improvement accordingly. We hope those improvements meet your requirements.

Reviewer 4 Report

Comments and Suggestions for Authors

Long-term adverse effects of chemotherapy and radiotherapy on the oral health of childhood cancer survivors: a systematic review and meta-analysis

This interesting article discusses long-term oral complications in children cancer survivors after chemotherapy and radiotherapy. The result is that children who have survived cancer have a higher risk of having dental changes than their control counterparts.

The manuscript, based on a systematic review and meta-analysis, is well written and constructed. The results are well presented, as is the discussion. The article also provides the strengths and limitations of this study.

I would suggest writing in Table 1 to use ChemoT or Chemotherapy to homogenise the text.

It would be nice to add some clinical pictures of the changes for non-specialist readers.

Author Response

We are pleased with the opportunity to revise and submit our manuscript entitled “Chemotherapy and radiotherapy long-term adverse effects on oral health 
of childhood cancer survivors: a systematic review and meta-analysis” (Manuscript ID - cancers-2776218).

Considering the editor and reviewers’ comments, all have been considered very important and were taken into profound consideration.

Manuscript changes are highlighted in the revised manuscript. Our point-by-point responses to all comments are outlined and detailed below. We hope that you find our responses satisfying.

We hope the revised manuscript will enable its further consideration. We are happy to consider further revisions and we thank you for your continued interest in our research.

Dear Reviewer

Thank you for all your valid comments, I hope I have clarified it in the best possible way, and I am available for any other questions that arise.

  1. I would suggest writing in Table 1 to use ChemoT or Chemotherapy to homogenise the text.

Our answer: We appreciate this valid question. We homogenized the terms in the text to ChemoT

  1. It would be nice to add some clinical pictures of the changes for non-specialist readers.

Our answer: Although we would be keen to have clinical pictures, such practice is uncommon in systematic reviews and may confuse readers. The use of clinical cases and/or case series is, at a wider extent, counter-indicated in this type of evidence-based studies, as per the Cochrane Handbook (https://handbook-5-1.cochrane.org/chapter_14/14_6_3_case_reports.htm), as they bring serious bias potential.

Round 2

Reviewer 2 Report

Comments and Suggestions for Authors

It is important for the authors to provide a synthesis of their work, and graphics are crucial for the article's readers. Including 15 figures is excessive; a synthesis should be created, and only the most essential ones should be incorporated into the main text, reducing the number to provide the reader with an overall view.

The identification of publication bias is typically done using funnel plots, which are scatter plots depicting the relationship between a measure related to standard error (sample size) and the results of each included trial, represented as points or circles around the meta-analysis result.

Furthermore, the consideration of heterogeneity is essential. Heterogeneity refers to the extent to which individual results from trials included in the meta-analysis differ from one another. When heterogeneity is high, this variability introduces uncertainty regarding whether the meta-analysis result accurately represents the true value within the population. It suggests that the meta-analysis may have included different research questions. Therefore, assessing heterogeneity is a fundamental aspect when evaluating the validity of the meta-analysis result.

Author Response

Reviewer

It is important for the authors to provide a synthesis of their work, and graphics are crucial for the article's readers. Including 15 figures is excessive; a synthesis should be created, and only the most essential ones should be incorporated into the main text, reducing the number to provide the reader with an overall view.

Our Answer: Although we agree that such number is excessive, those new figures were submitted as supplementary files to the sake of manuscript’s readibility, and also following your recommendation of adding them.

The identification of publication bias is typically done using funnel plots, which are scatter plots depicting the relationship between a measure related to standard error (sample size) and the results of each included trial, represented as points or circles around the meta-analysis result.

Our Answer: Publication bias is always based on statistical tests, with Egger test being one of the most recognized and used. Graphical interpretations of publication bias always come with subjectivity, they are important, but in the interest of the readers, we decided to just pinpoint the statistical significance of the multiple Egger tests used.

Furthermore, the consideration of heterogeneity is essential. Heterogeneity refers to the extent to which individual results from trials included in the meta-analysis differ from one another. When heterogeneity is high, this variability introduces uncertainty regarding whether the meta-analysis result accurately represents the true value within the population. It suggests that the meta-analysis may have included different research questions. Therefore, assessing heterogeneity is a fundamental aspect when evaluating the validity of the meta-analysis result.

Our Answer: We respectfully disagree with this commentary because we have evaluated sources of heterogeneity, such as table 5, that explored sensitivity analyses to that end.

Reviewer 3 Report

Comments and Suggestions for Authors

After a thorough review of the revised manuscript, I am pleased to note that the authors have addressed the previous concerns effectively. The improvements made in this version significantly enhance the quality and clarity of the work. The manuscript now meets the journal's standards and, in my opinion, can be accepted for publication.

Comments on the Quality of English Language

minor mistakes

Author Response

.